# Proteomic profiling of MIS-C patients indicates heterogeneity relating to interferon gamma dysregulation and vascular endothelial dysfunction

Caroline Diorio[1,2], Rawan Shraim[1,3], Laura A. Vella[4,5], Josephine R. Giles[5,6], Amy E. Baxter [5,6], Derek A. Oldridge[5,6,7], Scott W. Canna[2,8], Sarah E. Henrickson[9], Kevin O. McNerney[1], Frances Balamuth[10], Chakkapong Burudpakdee[1,2], Jessica Lee[1,2], Tomas Leng [1], Alvin Farrel[1,3], Michele P. Lambert [2,11], Kathleen E. Sullivan[2,9], E. John Wherry [5,6], David T. Teachey [1,2,12✉], Hamid Bassiri[2,4,12] & Edward M. Behrens[2,8,12]

Multi-system Inflammatory Syndrome in Children (MIS-C) is a major complication of Severe Acute Respiratory Syndrome Coronavirus 2 (SARS-CoV-2) infection in pediatric patients. Weeks after an often mild or asymptomatic initial infection with SARS-CoV-2 children may present with a severe shock-like picture and marked inflammation. Children with MIS-C present with varying degrees of cardiovascular and hyperinflammatory symptoms. Here we perform a comprehensive analysis of the plasma proteome of more than 1400 proteins in children with SARS-CoV-2. We hypothesize that the proteome would reflect heterogeneity in hyperinflammation and vascular injury, and further identify pathogenic mediators of disease. We show that protein signatures demonstrate overlap between MIS-C, and the inflammatory syndromes macrophage activation syndrome (MAS) and thrombotic microangiopathy (TMA). We demonstrate that PLA2G2A is an important marker of MIS-C that associates with TMA. We find that IFNγ responses are dysregulated in MIS-C patients, and that IFNγ levels delineate clinical heterogeneity.

[1] Division of Oncology, Department of Pediatrics, Children's Hospital of Philadelphia, University of Pennsylvania Perelman School of Medicine, Philadelphia, PA, USA. [2] Immune Dysregulation Frontier Program, Department of Pediatrics, Children's Hospital of Philadelphia, University of Pennsylvania Perelman School of Medicine, Philadelphia, PA, USA. [3] Department of Biomedical and Health Informatics, Children's Hospital of Philadelphia, University of Pennsylvania Perelman School of Medicine, Philadelphia, PA, USA. [4] Division of Infectious Diseases, Department of Pediatrics, Children's Hospital of Philadelphia, University of Pennsylvania Perelman School of Medicine, Philadelphia, PA, USA. [5] Institute for Immunology, University of Pennsylvania Perelman School of Medicine, Philadelphia, PA, USA. [6] Department of Systems Pharmacology and Translational Therapeutics, University of Pennsylvania, Philadelphia, PA, USA. [7] Division of Pathology and Laboratory Medicine, Department of Pediatrics, Children's Hospital of Philadelphia, University of Pennsylvania Perelman School of Medicine, Philadelphia, PA, USA. [8] Division of Rheumatology, Department of Pediatrics, Children's Hospital of Philadelphia, University of Pennsylvania Perelman School of Medicine, Philadelphia, PA, USA. [9] Division of Allergy and Immunology, Department of Pediatrics, Children's Hospital of Philadelphia, University of Pennsylvania Perelman School of Medicine, Philadelphia, PA, USA. [10] Division of Emergency Medicine, Department of Pediatrics, Children's Hospital of Philadelphia, University of Pennsylvania Perelman School of Medicine, Philadelphia, PA, USA. [11] Division of Hematology, Children's Hospital of Philadelphia, University of Pennsylvania Perelman School of Medicine, Philadelphia, PA, USA. [12]These authors contributed equally: David T. Teachey, Hamid Bassiri, Edward M. Behrens. ✉email: TEACHEYD@chop.edu

Multisystem inflammatory syndrome in children (MIS-C) emerged as the major pediatric complication of the Severe Acute Respiratory Syndrome Corona Virus 2 (SARS-CoV-2) pandemic of 2020[1–3]. MIS-C is characterized by fever and inflammation, and has some clinical features overlapping with Kawasaki disease (KD)[3–7]. Due to the similarities with KD, initial treatment regimens for MIS-C included both intravenous immunoglobulin (IVIG) and corticosteroids[8–10]. Data from randomized controlled trials comparing these two treatments is not yet available; however, best evidence to date suggests that IVIG alone provides suboptimal treatment for MIS-C[8].

Although the phenotypes of KD and MIS-C are overlapping, they have important clinical differences and immune profiling has delineated separate but overlapping signatures between the conditions[5,11–13]. Shock is a marked feature in patients with MIS-C, with the majority of patients requiring pediatric intensive care unit (PICU) admission[4]. The severity of MIS-C has invoked cytokine storm as a potential driver of symptoms[11]. SARS-CoV-2 infection leading to severe Coronavirus Disease (COVID-19) can also lead to cytokine storm[14]. Multiple cytokines including interferon gamma (IFNγ), interleukin-6 (IL-6), IL-8, IL-1, IL-17, and IL-1β have been implicated as potentially causative in patients with MIS-C[11,15]. The putative activity of cytokine dysregulation in the pathophysiology of MIS-C has led to the use of cytokine blockade with agents such as tocilizumab or anakinra in refractory patients; however, the optimal targeted agent has not yet been determined[9]. Improved understanding of the cytokine dysregulation in these patients is critical to identifying rational targeted therapies.

In addition to shock and inflammation, MIS-C and SARS-CoV-2 infection are associated with an increased risk of thrombosis in both adult and pediatric patients (Whitworth et al. *Blood*, in press)[16]. We have previously demonstrated that infection with SARS-CoV-2 and MIS-C are associated with thrombotic microangiopathy (TMA), and the associated biomarker soluble C5B9 (SC5B9)[17]. TMA is a group of diseases characterized by microangiopathic hemolytic anemia, thrombocytopenia, and organ dysfunction related to microthrombi[18]. TMA can occur as a primary process or secondary to infection, inflammatory insult or in hematopoietic stem cell transplant (HSCT)[18,19]. The mechanism of TMA in the context of MIS-C is not known, and its association with cytokine dysregulation remains unclear.

In this work we interrogate >1400 proteins in the plasma proteome and integrate this information with clinical and high-dimensional flow cytometry data. We hypothesize that distinct dysregulations in proteins and associated pathways that relate to cytokine storm and vascular injury contribute to the underlying pathophysiology of MIS-C. We demonstrate dysregulation of the interferon gamma pathway as a major contributor to MIS-C pathophysiology, and identify candidate biomarkers and predictors of disease severity.

## Results

**Patients included in the study.** Between April 2020 and October 2020, we enrolled 63 hospitalized patients with MIS-C (N = 22), Severe COVID-19 (Severe, N = 15) or asymptomatic or mild SARS-CoV-2 infection (Minimal, N = 26). In addition, we included remnant plasma samples from otherwise healthy patients (Healthy, N = 25). Details of the clinical presentation of MIS-C, Severe, and Minimal patients are presented in Supplementary Tables 1 and 2. Like previously reported cohorts, a high proportion (N = 17, 77%) of patients with MIS-C were admitted to the PICU[2,4]. We utilized the Olink Explore 1536/384 protein biomarker platform to interrogate the plasma proteome of all 88

patients. A subset of patients included in this study have been reported previously[17,20–23]. Flow cytometry data was available for 20 patients included in this study (MIS-C N = 8, Severe N = 6, Minimal N = 6), reported previously by Vella et al.[20].

We first validated the accuracy of the O-link platform. Patients with MIS-C in our sample had marked elevations of brain-type natriuretic peptide (BNP) and troponin measured clinically (Supplementary Table 2). The Olink panel measures N-terminal prohormone brain natriuretic peptide (NTproBNP), a protein pre-cursor to BNP associated with cardiac damage[24]. NTproBNP was significantly higher in MIS-C patients (Supplementary Fig. 1a)[15] and Olink NTproBNP correlated highly with clinical BNP (R = 0.78, p = 2.7e−08; Supplementary Fig. 1b).

We compared cytokines measured in our laboratory with the same cytokines measured by Olink and demonstrated strong correlations (Supplementary Fig. 1c). We and others have previously found elevations in IL-10 distinguish patients with MIS-C from those with acute SARS-CoV-2 infection. We confirmed this observation with data from Olink (Supplementary Fig. 1d)[13,21].

**Overall architecture of pediatric SARS-CoV-2 and MIS-C plasma proteome.** Following this validation, we took an unbiased approach to investigate the proteomic data. Overall architecture of the data is presented in Fig. 1. Patients with MIS-C cluster differently from patients with acute SARS-CoV-2 or healthy patients when visualized with t-distributed stochastic neighbor embedding (tSNE; Fig. 1a). Vella et al. previously demonstrated that MIS-C is associated with CD8+ T-cell activation and Tbet+ plasmablasts[20]. In the subset of patients on whom flow cytometry data was available, we overlaid flow cytometry markers on tSNE clustering. Interestingly, there was heterogeneity in percent of activated CD8+ T cells and percent of Tbet+ plasmablasts within the MIS-C clusters (Fig. 1a)[20].

To look at the most prominent drivers of difference between groups, we used principal component analysis (PCA) to visualize clustering between patients. To achieve better separation between groups, we limited the analysis to proteins that were differentially expressed between any of the four groups (N = 231). Lists of differentially expressed proteins (DEP) between all groups are available in Supplementary Data 1. Healthy patients clustered separately from those with Severe, Minimal or MIS-C (Supplementary Fig. 2a), while patients with Severe and MIS-C overlapped. The corresponding Scree plot and top 20 contributions to each dimension of the PCA are available in Supplementary Fig. 2b–d. Multiple IFNγ responsive proteins contribute to PCA Dimension 2, including IL-27, CXCL9, IL18BP, and CCL23.

Twenty patients with MIS-C received treatment with both steroids and IVIG. One patient was treated with IVIG alone, and one patient did not receive treatment. Convalescent samples were available on 12 patients (54%). To understand how the overall proteome of patients with MIS-C changes over time, we first created a PCA mapping all proteins and all patients (Supplementary Fig. 2e) and then PCA transformed data for convalescent samples on that space (Supplementary Fig. 2f). Convalescent samples shifted towards the healthy controls, implying that following treatment, the proteome in MIS-C patients returns towards a baseline state.

Pre-treatment and post-treatment samples were available on 5 patients. To investigate if the timing of initial collection relative to treatment altered the proteome, we assessed change over time in one of the most differentially expressed proteins, phospholipase A2 (PLA2G2A; Supplementary Fig. 1e). PLA2G2A pre- and post-treatment samples did not appear to differ significantly when measured in the acute time frame suggesting that sampling of

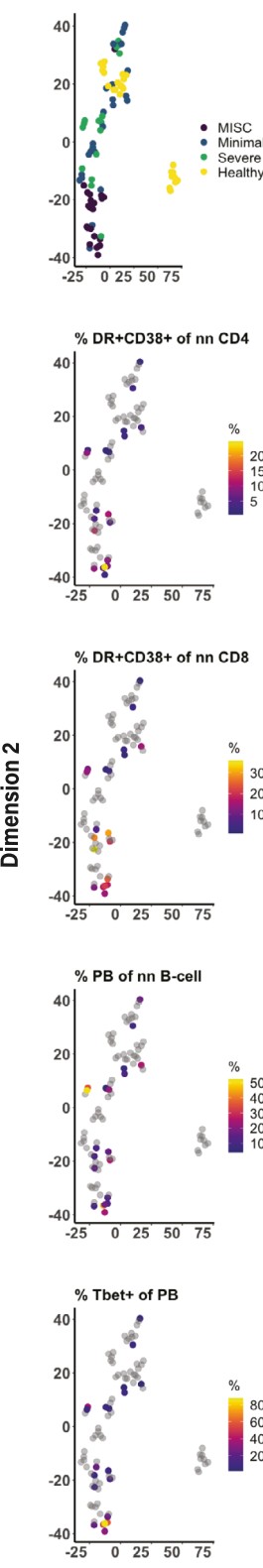

Fig. 1 Overarching architecture of plasma proteome in patients with MIS-C ($N = 22$), minimal SARS-CoV-2 ($N = 26$) infection, and Severe COVID-19 ($N = 15$) compared to healthy controls ($N = 25$). t-distributed Stochastic Neighbor Embedding (tSNE) plots are used to visualize clustering between the four groups of patients with overlay from flow cytometry-based scores of percent of non-naïve CD4+ T cells that are activated (HLA-DR+CD38+), percent non-naïve CD8+ T cells that are activated, percent of non-naïve B cells that are plasmablasts, and percent of plasmablasts that are T-bet+. Gray dots indicate that data was not available. Source data are provided as a Source Data file.

**Characterizing pathways of activation in patients with MIS-C and SARS-CoV-2 infection.** We used unbiased exploration of DEPs to understand differences between MIS-C, Minimal and Severe patients compared to healthy patients. Volcano plots are presented in Fig. 2. Notably, PLA2G2A is highly differentially expressed between all three groups compared to healthy patients, with the most marked difference occurring in MIS-C patients.

To identify pathways of relevance to the pathophysiology of MIS-C and pediatric COVID-19 we performed pathway analysis accounting for protein–protein interactions using pathfindR[25]. Ranked pathways and enrichments are presented in Fig. 2a–c. Lists of all ranked pathways are available in Supplementary Data 2. A protein to pathway analysis on the pathways of interest in MIS-C was also performed (Fig. 2a).

Cytokine–cytokine and chemokine–cytokine receptor pathways were dysregulated in all classifications of pediatric SARS-CoV-2 infection. Notably, similar pathways were perturbed in MIS-C, Minimal and Severe patients compared to healthy controls. Overlaps of DEPs between disease states are shown in Supplementary Fig. 3a. The majority of DEPs in Severe COVID-19 are also differentially expressed in MIS-C. We directly assessed the DEPs between these groups and surprisingly, there were very few differences (Supplementary Fig. 3b). As this comparison involved a smaller proportion of patients, we used a nominal p-value with a threshold of 0.05.

**MIS-C is characterized by a disproportionate response to interferon gamma.** IFNγ has been associated with MIS-C in previous reports[12,15]. We previously described that IFNγ levels were high in patients with Severe and MIS-C, that levels of IFNγ could not distinguish between these two groups and that IL-10 can distinguish MIS-C from Severe (Supplementary Fig. 3b)[21]. In order to better understand these two cytokines, we investigated correlations between IFNγ and IL-10 and their canonical response proteins and proteins correlates of cellular populations responsible for their production (Fig. 3). We examined CXCL9 as the key protein associated with IFNγ response and found that all patients with SARS-CoV-2 infection showed a positive correlation between CXCL9 and IFNγ. Strikingly, MIS-C patients had a disproportionately high CXCL9 response to IFNγ compared to the other groups (Fig. 3a; $p < 0.001$ by logistical regression modeling). To probe the source of IFNγ production, we correlated IFNγ levels with soluble markers of activated T cells (IL2RA), NK-cells (NCR1), and macrophages (CD163). IFNγ levels in MIS-C patients associated significantly with IL2RA and NCR1 but not CD163, suggesting that IFNγ levels correlate with T cells and NK-cells but not macrophages (Fig. 3a).

Consistent with prior reports, levels of IL-10 are significantly higher in patients with MIS-C than in those with Severe COVID-19 (Supplementary Fig. 1d)[13,21]. Similar to the analysis above, we examined the expression of IL12B, which should be inhibited by IL-10[26]. IL12B was inversely correlated with IL-10 in patients with MIS-C, but not in Minimal, Severe or Healthy patients (Fig. 3b). This

plasma shortly after treatment does not immediately affect this analyte. Levels of PLA2G2A and IFNγ (Supplementary Fig. 1e, f) did eventually decay over time following treatment. We also examined the impact of treatment in all proteins in Supplementary Fig. 2f. Samples drawn prior to or immediately after receiving treatment did not cluster separately from each other, implying minimal differences in the short-term following treatment.

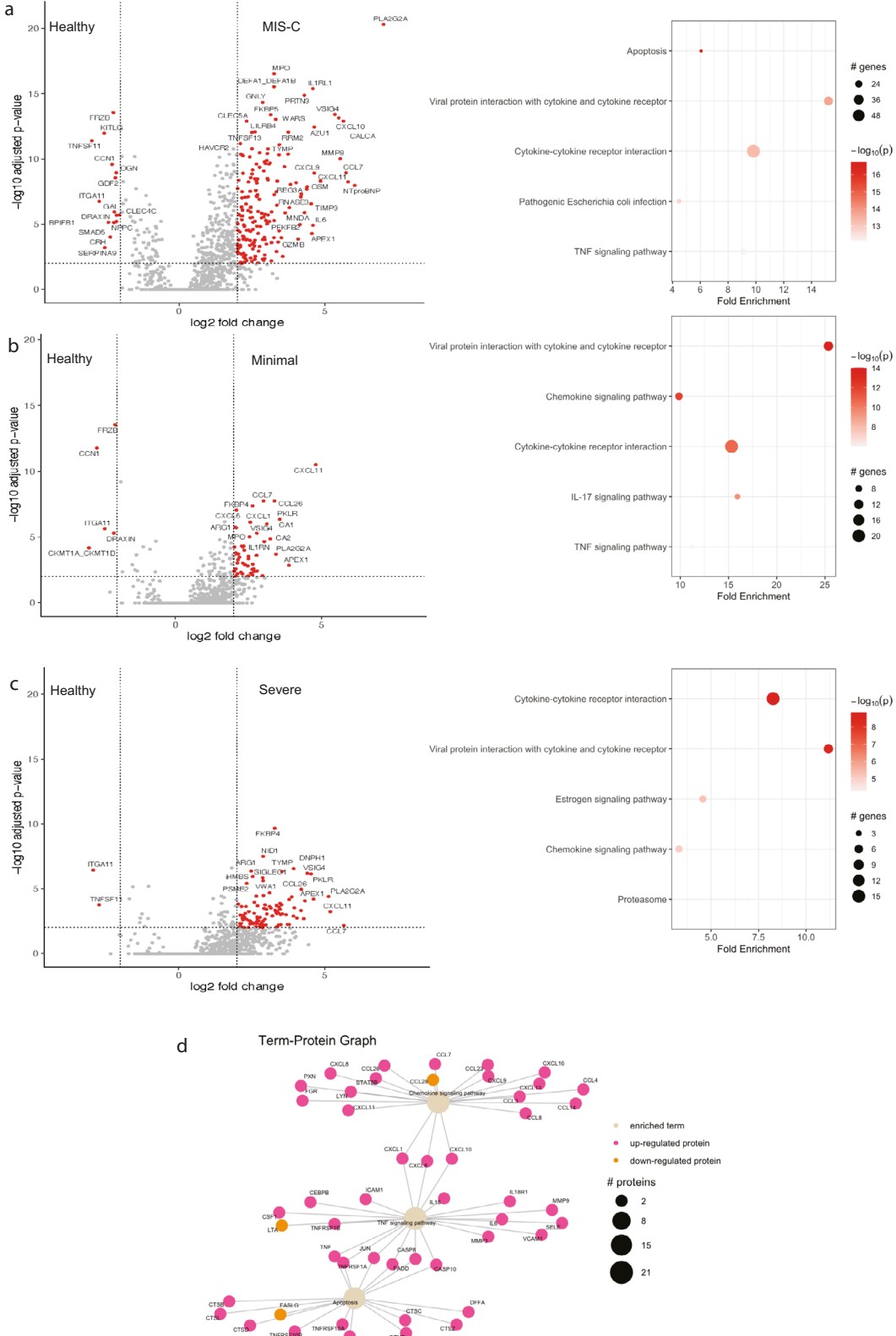

**Fig. 2 Differentially expressed proteins and ranked pathway analysis for each disease state compared to healthy controls.** Differentially expressed proteins (DEPs), log2 fold change threshold of 2 and FDR threshold of 0.01, between patients with MIS-C ($N = 22$) and healthy controls ($N = 25$) are shown in **a**, along with ranked pathway analysis. Size of dots represents number of enriched genes and intensity of colors represents *p*-value associated with enrichment analysis. DEPs and ranked pathway analysis are shown for patients with minimal Severe Acute Respiratory Syndrome Corona Virus 2 (SARS-CoV-2; $N = 26$) infection compared healthy controls ($N = 25$) (**b**), and Severe Coronavirus Disease (COVID-19; $N = 15$) patients compared to healthy controls ($N = 25$) (**c**). Network analyses for ranked pathway analysis between patients with MIS-C ($N = 22$) and healthy controls are shown in **d** where up-regulated proteins are colored in green and down-regulated proteins are colored in red. Node size of the pathway term represents number of input proteins implicated in the pathway. Source data are provided as a Source Data file.

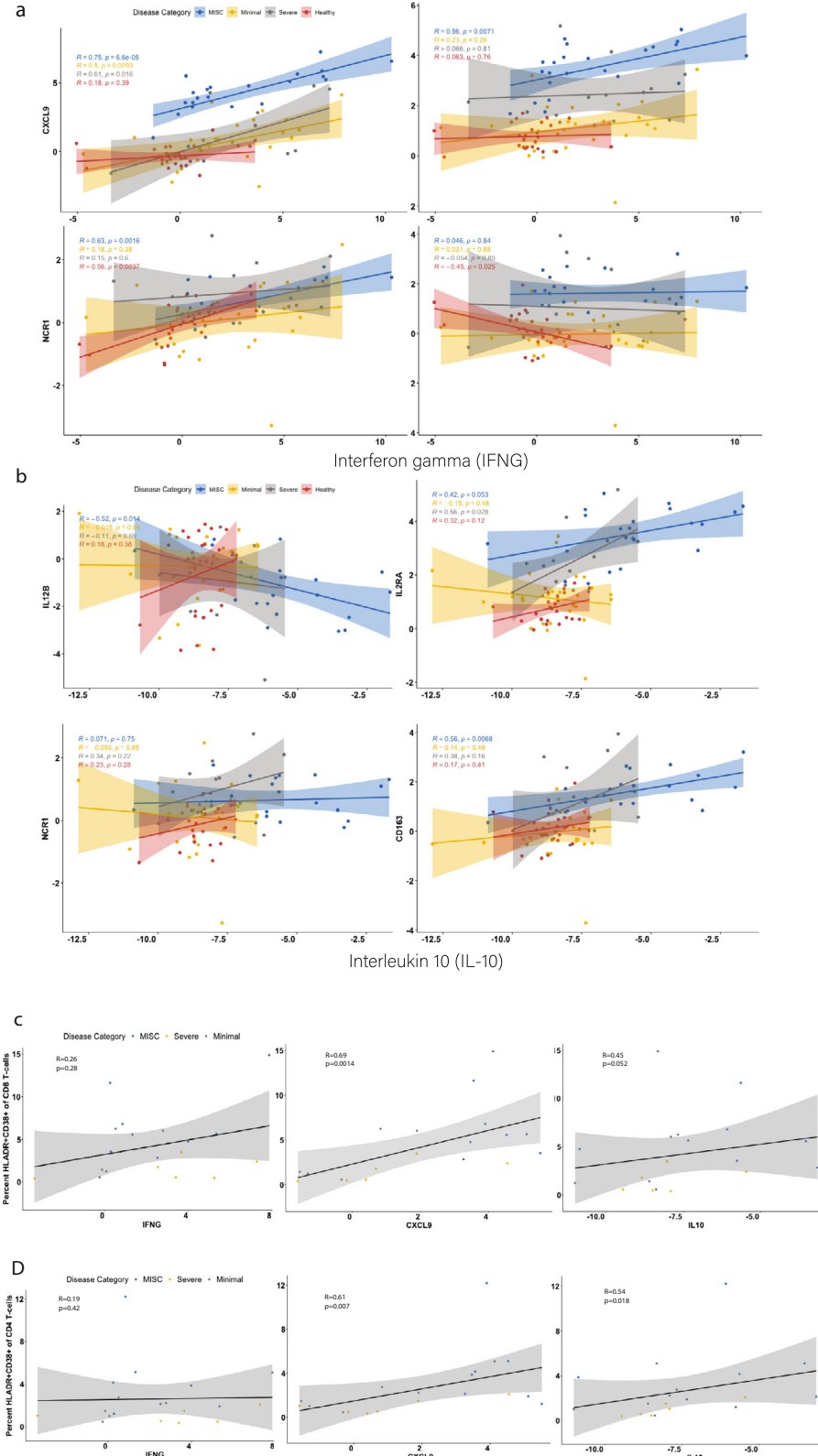

finding implies that the elevated IL-10 in MIS-C patients is bioactive. IL-10 production was significantly associated with markers of activated T cells and macrophages, but not with NK-cells (Fig. 3b).

We subsequently examined the relationship between IFNγ, CXCL9, and IL-10 and activated CD8⁺ and CD4⁺ T cells from flow cytometry data to probe these correlations further (Fig. 3c, d). We found that CXCL9 was significantly and strongly correlated

with both cell types but IFNγ was not. IL-10 was significantly associated with activated CD4⁺ but not activated CD8⁺ T cells.

**MIS-C patients have an MAS-like phenotype characterized by IFNγ and CXCL9 signaling.** The outsized response to IFNγ with excessive CXCL9 expression in patients with MIS-C evoked

**Fig. 3 IFNγ and IL-10 responses and associated cell types.** Spearman correlations between IFNγ signaling and its canonical response protein (CXCL9), as well as proteins associated with activated T cells (IL2RA), NK cells (NCR1), and macrophages (CD163) for patients with (MIS-C; $N = 22$), Minimal Severe Acute Respiratory Syndrome Corona Virus 2 infection (SARS-CoV-2; $N = 26$) infection, and Severe Corona Virus Disease (COVID-19; $N = 15$) compared to healthy controls ($N = 25$) are shown in panel **a**. Correlations between IL-10 and its canonical receptor IL12B, as well as IL2RA, NCR1, and CD163 for each disease category are shown in panel **b**. Panel **c** shows correlations between HLADR+CD38+ CD8+ T cells ($N = 19$) and IFNγ, CXCL9, and IL-10. Dots are colored by each patient's disease category. Panel **d** demonstrates correlations between HLADR+CD38+ CD4+ T cells ($N = 19$) and IFNγ, CXCL9, and IL-10. Dots are colored by disease category. Error bars represent 95% confidence interval. All p-values were calculated with a two-sided test. Source data are provided as a Source Data file.

macrophage activation syndrome (MAS)[27,28]. MAS is a hyper-inflammatory condition that occurs in the setting of rheumato-logic diseases, infections or as a primary disorder, characterized by excessive activation of lymphocytes and macrophages in the setting of marked IFNγ production[29,30]. We investigated whether IFNγ, CXCL9, and other markers associated with MAS could distinguish different SARS-CoV-2 outcomes (Fig. 4a)[28,30,31]. Unsupervised hierarchical clustering showed two distinct groups of MIS-C patients that had elevations in most MAS markers. These two clusters primarily differed by their IFNγ expression with an IFNγ-high versus IFNγ-low group evident.

To further interrogate the hypothesis that MAS was associated with MIS-C, we applied a modified version of the Ravelli MAS criteria to all patients on whom ferritin had been measured ($N = 40$; MIS-C $N = 21$, Minimal $N = 6$, Severe $N = 13$)[32]. If parameters were missing, they were imputed as negative to bias our score towards the null hypothesis. We compared patients who met criteria for MAS to those who did not (Supplementary Table 3). CD163, CXCL9, IFNγ, IL2RA, and VSIG4 were significantly elevated in patients who met MAS criteria (Fig. 4b). We found that maximum ferritin during admission was significantly higher in patients that met criteria for MAS than those that did not ($P < 0.0001$) but did not differ significantly between the disease groups (Fig. 4c, d). In MIS-C patients on whom both an acute and a convalescent sample was available ($N = 12$), IFNγ, CXCL9, and IL2RA all significantly decreased over time (Fig. 4e).

MAS is typically associated with cytopenias, including neutropenia. In contrast, MIS-C is characterized by marked neutrophilia[4], and in patients who met criteria for MAS high absolute neutrophil counts (ANC) were apparent (Fig. 4f). ANC was significantly higher in patients with MIS-C than in those with Severe or Minimal (Fig. 4g). In MIS-C patients, neutrophilia did not correlate with IFNγ or CXCL9 but correlated strongly with CD163, implying neutrophilia in these patients is associated with macrophage activation ($R = 0.72$, $p < 0.0001$; Fig. 4h), but not directly related to IFNγ or CXCL9. Macrophage hyperrespon-siveness to IFNγ expression has been linked to TRIM8 dysregulation in MAS[33]. We noted dysregulation of another TRIM protein, the IFNγ signaling suppressive protein TRIM21[34,35]. Consistent with a derepression of IFNγ signaling, MIS-C patients expressed less TRIM21 compared to healthy controls (Supplementary Fig. 3c).

**PLA2G2A is a candidate biomarker for MIS-C and is associated with a thrombotic microangiopathy phenotype.** We and others have previously demonstrated that infection with SARS-CoV-2 is associated with clinical TMA and elevations in the TMA biomarker SC5b9[17,36]. As SC5b9 was not included in the Olink analysis, we correlated SC5b9 levels measured in our laboratory ($N = 75$) to Olink proteins related to vascular or platelet dys-function to identify surrogate biomarkers. Hierarchical clustering identified that SC5b9 correlated most highly with PLA2G2A, PDGFC, SELE, CALCA, NOS3, VWA1, and TYMP (Fig. 5a).

We evaluated whether proteins highly correlated with SC5b9 were able to identify disease category (Fig. 5b). Patients with MIS-C clustered due to very high PLA2G2A expression, and moderate CALCA and TYMP expression. PLA2G2A and CALCA were both significantly higher between MIS-C and Severe patients (Fig. 5c). PLA2G2A is significantly higher in SARS-CoV-2 infected patients than in healthy patients with the most marked difference in patients with MIS-C (Fig. 5c). In MIS-C patients, PLA2G2A, SELE, CALCA, and VWA1 all improved between the acute and convalescent phases (Fig. 5d). Notably, levels of PLA2G2A do not return completely to normal during convalescence.

For patients who had blood smears available for review ($N = 34$; MIS-C $N = 15$, Minimal $N = 11$, Severe $N = 8$) we applied simple criteria for TMA, as previously published (Supplementary Table 4)[17]. Meeting criteria for TMA was associated with a significantly higher PLA2G2A and CALCA (Fig. 5e). Patients who met criteria for TMA were more likely to require inotropes during their admission than those who did not meet criteria for TMA (92% vs 23% respectively, $N = 34$, Wilcoxon test p-value <0.001). Given our hypothesis that PLA2G2A is a marker of TMA and microangiopathic hemolytic anemia, we examined whether levels of PLA2G2A correlated to lowest platelet count and lowest hemoglobin during admission. PLA2G2A levels inversely corre-lated with platelets and hemoglobin with groupings by disease category evident (Fig. 5f).

**Clinical heterogeneity among MIS-C patients.** Next, we inves-tigated the intersection between the MAS and TMA phenotypes and MIS-C. In all patients who had available data for analysis of MAS and TMA, we looked at overlap with MIS-C ($N = 21$). Among MIS-C patients ($N = 15$), TMA and MAS occurred independently (Fisher's exact, $p = $ n.s.; Fig. 6a), although the sample size was small. When TMA and MAS phenotypes were overlayed on the same tSNE mapping used in Fig. 1, patient clustering by disease state was evident (Supplementary Fig. 4a).

To further explore the heterogeneity among MIS-C patients, we compared patients from the IFNγ-high ($N = 7$) and IFNγ-low ($N = 13$) clusters identified in Fig. 4a. We looked at DEPs between these two clusters and performed unbiased pathway and clustering analysis (Fig. 6b–d) indicating differences in cytokine signaling including IL-17 pathways. IL-17 is a neutrophil chemotactic factor and may relate to the neutrophilia present in these patients[37]. In Supplementary Fig. 4b IFNγ levels are overlayed on the same tSNE map as displayed in Fig. 1, demonstrating heterogenous clustering within MIS-C.

We examined occurrence of TMA and MAS among these IFNγ-high and -low MIS-C clusters. There was no increase in TMA among IFNγ-high MIS-C patients (Fisher's exact, $p = $ n.s.; Supplementary Table 5). There was a trend towards a significantly increased enrichment in MAS among IFNγ-high MIS-C patients compared to IFNγ-low patients (83% vs. 38%, Fisher's exact $p = 0.14$; Supple-mentary Table 5). In keeping with this trend, IFNγ-high MIS-C patients had significantly higher ferritin (Wilcoxon Rank-Sum Test, $p = 0.036$; Fig. 6e). IFNγ status did not associate with fibrinogen, d-dimer, platelet, hemoglobin or ANC.

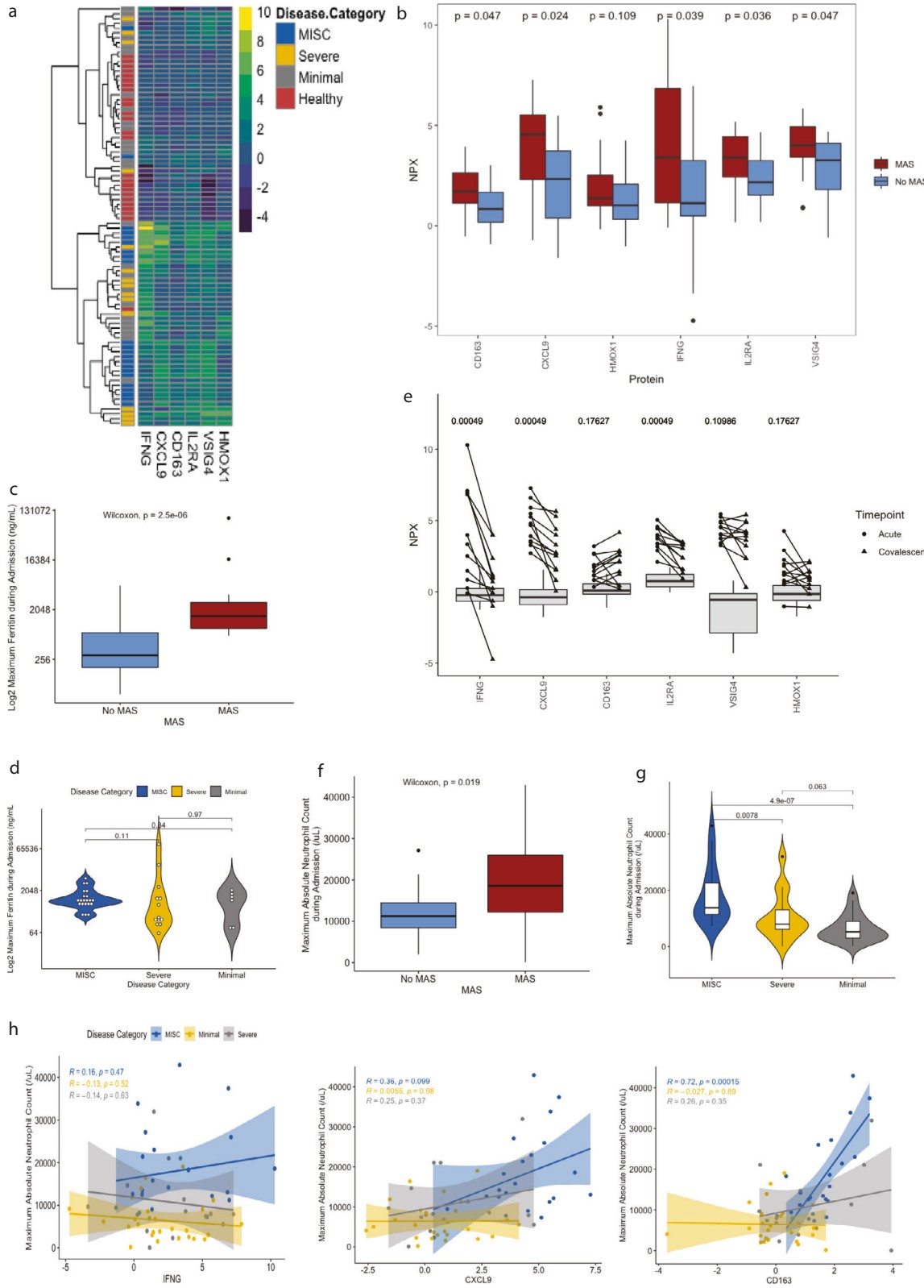

To understand the relationship between IFNγ and clinical outcomes in MIS-C, we used requirement for inotropes as a surrogate markers of severity. Surprisingly, patients within the IFNγ-high cluster had a lower likelihood of requiring inotropes than those in the IFNγ-low cluster (Fig. 6f), implying that higher levels of IFNγ are associated with less severe illness in MIS-C patients. NTproBNP, a marker associated with cardiac

damage, showed a similar trend, although the result was not statistically significant, likely due to a lack of power (Fig. 6g). We performed this analysis on all patients with MIS-C and found that higher levels of NTproBNP were higher in patients requiring inotropes (Fig. 6h). Thus, it appears higher levels of IFNγ may be associated with less severe illness in patients with MIS-C.

**Fig. 4 Association with MAS and patients with MIS-C and SARS-CoV-2 infection. a** Heatmap of MAS-associated proteins IFNγ, CXCL9, CD163, IL2RA, VSIG4, and HMOX1 with unsupervised hierarchical clustering applied to all patients ($N = 88$). Color bar represents range of variable NPX for each protein. **b** Boxplot demonstrating levels of MAS-associated proteins in patients who met ($N = 17$) and did not meet modified criteria for MAS ($N = 23$). *P*-values computed with Wilcoxon test. Horizontal line represents median, with bounds of box representing interquartile range. Whiskers represent 1.5x the interquartile range. Dots represent outliers. **c** Median log2 transformed maximum ferritin during admission in patients who did ($N = 17$) and did not ($N = 23$) meet criteria for MAS. $P = 0.0000025$, *p*-value computed with Wilcoxon test. **d** log2 transformed maximum ferritin during admission for patients with MIS-C ($N = 21$), Severe COVID-19 ($N = 13$), and minimal disease ($N = 6$). *P*-values computed with pairwise comparisons using Wilcoxon rank sum test following Kruskal–Wallis testing. **e** Acute and convalescent levels of each MAS-related protein for MIS-C patients on whom both acute and convalescent samples were available ($N = 12$). Circles represent acute samples with triangles representing convalescent samples. Lines connect matched pairs. Gray boxes and whiskers show median and interquartile range of healthy controls ($N = 25$). *P*-values for difference between acute and convalescent samples were computed using a paired samples Wilcoxon test. Horizontal line represents median, with bounds of box representing interquartile range. Whiskers represent 1.5x the interquartile range. Dots represent outliers. **f** Boxplot of maximum absolute neutrophil count during admission for patients who met ($N = 17$) and did not ($N = 23$) criteria for MAS. *P*-value computed with Wilcoxon test. Horizontal line represents median, with bounds of box representing interquartile range. Whiskers represent 1.5x the interquartile range. Dots represent outliers. **g** Maximum absolute neutrophil count during admission for patients with MIS-C ($N = 22$), Severe disease ($N = 15$), and Minimal disease ($N = 26$). *P*-values computed with pairwise comparisons using Wilcoxon rank sum test following Kruskal–Wallis testing. Horizontal line represents median, with bounds of box representing interquartile range. Whiskers represent 1.5x the interquartile range. Dots represent outliers. **h** Maximum absolute neutrophil count during admission correlated with IFNγ, CXCL9, and CD163. R computed using Spearman correlation. Error bands represent 95% confidence interval. All *p*-values were calculated using two-sided tests. Source data are provided as a Source Data file.

Previously, Vella et al. demonstrated that inotrope need was associated with higher frequencies of activation of CX3CR1+ CD8+ T cells[20]. CX3CR1 is expressed on CD4+ T cells, monocytes, and effector-like CD8+ T cells, and allows cells to interact with CX3CL1-expressing vasculature[38,39]. In MIS-C patients, activation of CX3CR1+ CD8+ T cells (but not CX3CR1+ CD4+ T cells; Supplementary Fig. 4c) and IFNγ inversely correlated (Fig. 6i; $N = 7$, Pearson $R = -0.83$, $p = 0.02$). However, levels of the CX3CR1 ligand, CX3CL1 had no relationship to CX3CR1+ CD8+ T-cell activation (Fig. 6i). In our unbiased network analysis (Fig. 6d) we identified CX3CL1 as an enriched protein despite it not reaching the thresholds used for our differential expression analysis (Fig. 6c). When explicitly tested, we surprisingly observed an association between elevated CX3CL1 and IFNγ-high patients, a group of patients we demonstrated to be at lower risk for inotrope use (Fig. 6j). The data suggest that free CX3CL1 levels in the plasma are independent of the frequency of CX3CR1+ CD8+ T-cell activation (Fig. 6i), explaining their differential association with inotrope requirement. Thus, we demonstrate that patients with high IFNγ levels are less likely to require inotrope use and accordingly have lower levels of CX3CR1+ CD8+ T-cell activation.

## Discussion
We used proteomic analysis to begin to unravel the complex and heterogeneous pathophysiology associated with MIS-C. We identify PLA2G2A as a candidate biomarker for MIS-C and show that PLA2G2A is associated with clinical features of TMA. We demonstrate that MIS-C patients are characterized by a disproportionately high CXCL9 response to IFNγ, implying a dysregulated response to IFNγ. We found that a subset of patients with MIS-C met modified criteria for the IFNγ-associated syndrome MAS. Patients with MIS-C showed heterogeneity based on IFNγ expression, and surprisingly, patients with higher IFNγ levels were less likely to require inotropes.

PLA2G2A is involved in host inflammatory responses and is associated with release of damage associated molecular pattern molecules (DAMPs) that activate the innate immune response[40,41]. PLA2G2A released from activated platelets is associated with neutrophilia[40]. Expression of PLA2G2A can be driven by IFNγ secretion[42–44]. In HSCT associated TMA, IFNγ has been identified as a key causative agent and part of an interferon-complement loop that contributes to vessel injury and

TA-TMA pathophysiology[45]. Overlapping phenotypes of MAS and TMA have been reported by others in other disease settings[46]. We hypothesize that patients with MIS-C may be responding to IFNγ with excessive PLA2G2A production. However, this relationship is likely complex as patients with high IFNγ expression appear to be protected from cardiac toxicity associated with MIS-C. Whether or not there is a causative relationship between PLA2G2A and TMA or MIS-C will need to be determined in future experiments. Understanding the mechanism of PLA2G2A in MIS-C pathophysiology is crucial because PLA2G2A and its associated pathways can be targeted with the common, inexpensive medication indomethacin[47]. Furthermore, higher levels of PLA2G2A associate with clinical findings of TMA. Regardless of the cause of PLA2G2A expression, the levels are markedly high in almost all patients with MIS-C and differentiate MIS-C patients form other SARS-CoV-2 infected patients, identifying PLA2G2A as a candidate biomarker for MIS-C. Future studies will need to prospectively examine PLA2G2A in patients with MIS-C and acute SARS-CoV-2 infection and validate its potential as a diagnostic biomarker of MIS-C.

Patients with MIS-C produced excessive CXCL9 in response to IFNγ. CXCL9 production has been associated with an increased risk of mortality in adult patients with severe COVID-19 disease and may contribute to cytokine storm in that setting[48]. One potential explanation for the heightened CXCL9 response to IFNγ in MIS-C patients is lower levels of TRIM21, a protein notable for its ability to repress IFNγ signaling through degradation of interferon response factors[34,35]. TRIM21 is induced by IFNγ, and indeed was one of the upregulated factors in the IFNγ-high MIS-C group, suggesting that this negative feedback loop is at least still partially intact in MIS-C despite the generally lower levels of TRIM21[49]. In keeping with IFNγ dysregulation, found that a subset of patients with MIS-C met modified criteria for MAS. We note that an IFNγ-high and -low subgrouping of MIS-C has been noted by others[7]. While the connection to MAS was also invoked by Esteve-Solie et al., we have clearly shown a connection between IFNγ-high MIS-C patients and objective MAS features in our data.

Other features of MAS in patients with MIS-C have been reported in the literature, including overlapping cytokine phenotypes and hemophagocytosis on bone marrow aspirates[50]. Although the evidence suggests that MIS-C does share many things in common with MAS, the clinical presentation of MIS-C patients is not fully consistent with a complete MAS phenotype

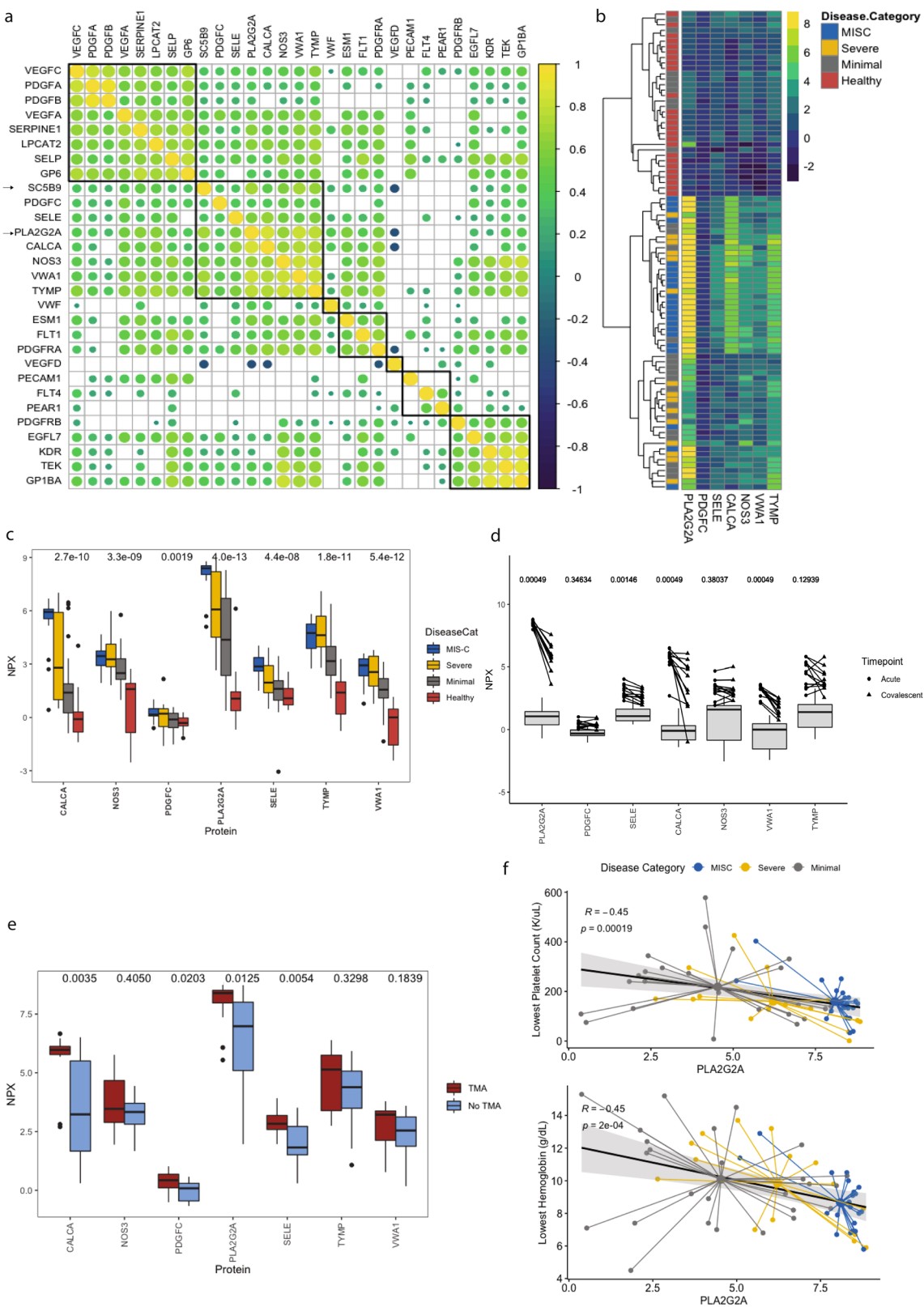

and may be better understood as an occult MAS[51,52]. MAS is often treated with IL-1β blockade[53]. Interestingly, IL-1β blockade with anakinra, a recombinant IL-1 receptor antagonist has been reported in many case reports as efficacious in MIS-C, and it is recommended for refractory patients in the American College of Rheumatology guidelines[54]. We further note that IL-1RN, the endogenous IL-1 receptor antagonist is upregulated in the IFNγ

high group (Fig. 6b). It is interesting to speculate whether this confers some of the cardiac protection seen in this group.

Curiously, we found that IFNγ-high patients were less likely to require inotropes, a result that is not consistent with other disorders associated with dysregulated IFNγ production[28]. This result was confirmed by an anti-correlation between IFNγ and CD8+ CX3CR1+ T cells, a cell population which has been

**Fig. 5 Evidence of thrombotic microangiopathy and vascular endothelial dysfunction in patients with MIS-C and SARS-CoV-2 infection. a** Correlation matrix of vascular and platelet related proteins and soluble C5B9 (SC5B9) in all patients on whom an SC5B9 was measured ($N = 75$). Hierarchical clustering was applied to identify surrogate markers associated with SC5B9. Color bar represents range of $R$ correlation between $-1$ and $+1$. **b** Heatmap of candidate surrogate markers associated with SC5B9 including PLA2G2A, PDGFC, SELE, CALCA, NOS3, VWA1, and TYMP. Unsupervised hierarchical clustering was applied to all patients ($N = 88$) with boxes colored by disease category. Color bar represents range of variable NPX for each protein. **c** Boxplots of each SC5B9 related protein across disease states (MIS-C $N = 22$, Severe $N = 15$, Minimal $N = 26$, Healthy $N = 25$). *P*-values computed by Kruskal–Wallis testing. Horizontal line represents median, with bounds of box representing interquartile range. Whiskers represent 1.5x the interquartile range. Dots represent outliers. **d** Acute and convalescent levels of each SC5B9-related protein for MIS-C patients on whom both acute and convalescent samples were available ($N = 12$). Circles represent acute samples with triangles representing convalescent samples. Lines connect matched pairs. Gray boxes and whiskers show median and interquartile range of healthy controls ($N = 25$). *P*-values for difference between acute and convalescent samples were computed using a paired samples Wilcoxon test. Horizontal line represents median, with bounds of box representing interquartile range. Whiskers represent 1.5x the interquartile range. Dots represent outliers. **e** Boxplot demonstrating levels of SC5B9-associated proteins in patients who met ($N = 13$) and did not meet criteria for thrombotic microangiopathy (TMA; $N = 21$). *P*-values computed with Wilcoxon test. Horizontal line represents median, with bounds of box representing interquartile range. Whiskers represent 1.5x the interquartile range. Dots represent outliers. **f** Correlations between lowest platelet count and lowest hemoglobin during admission and PLA2G2A. Small circles represent individual patients with large central circles representing the mean for each disease category. Dots are colored by disease category as MIS-C ($N = 22$), Severe ($N = 15$) or Minimal ($N = 26$). $R$ computed with Spearman correlation. Error band represents 95% confidence interval. All *p*-values were calculated using two-sided tests. Source data are provided as a Source Data file.

previously associated with an increased need for inotropes in MIS-C patients[20]. The observation that IFNγ may predict cardiac pathology in MIS-C may prove to be useful clinically as cytokine testing becomes more readily available in hospital settings. Future work should test this prospectively.

We have made several observations about the underlying pathophysiology of MIS-C; however, our study is limited by several factors. First, we retrospectively applied clinical categorizations for TMA and MAS to patients included in this study in order to understand relationships between clinical phenotypes and cytokine signatures. Neither of the clinical criteria used have been validated in this setting, and future studies should prospectively validate clinical criteria for these disorders in pediatric patients. Due to the rarity of this condition, we are underpowered to examine the full scope of clinical heterogeneity among MIS-C patients. We have identified important pathways to be examined in further detail in future mechanistic studies.

MIS-C is an important emergent syndrome characterized by excessive cytokine production and vascular-endothelial dysfunction. We have demonstrated that PLA2G2A is an important marker of MIS-C and associates with TMA. We have shown dysregulations in IFNγ responses in MIS-C patients and that IFNγ levels themselves indicate clinically relevant heterogeneity within MIS-C. Future studies should mechanistically examine the relationships between IFNγ, PLA2G2A, and their drivers in MIS-C initiation.

## Methods

**Study approvals.** This study was conducted in accordance with the Declaration of Helsinki and received approval from the Institutional Review Board (IRB) at CHOP. Verbal consent for this minimal risk study was obtained from patients or their legally authorized representative. Consent forms were signed by the consenting study team member and a copy was provided to the study participant or legally authorized representative. If appropriate, assent was obtained from children who were 7 years of age or older. Participants were not compensated for participation.

For remnant samples obtained from healthy controls, protected health information (PHI) was not recorded. A limited chart review of this cohort was granted by the CHOP IRB to determine that patients met criteria to be considered healthy. CHOP IRB granted exemption criteria per 45 CFR 46.104(d) 4(ii) and waiver of HIPPA authorization.

**Study design and population.** Patients were eligible to be prospectively enrolled in the Children's Hospital of Philadelphia (CHOP) SARS-CoV-2 biobank if they were admitted to CHOP during the COVID-19 pandemic and had either a positive SARS-CoV-2 reverse transcriptase polymerase chain reaction (RT-PCR) result from upper respiratory tract mucosa, had a positive SARS-CoV-2 antibody or met criteria for MIS-C. These enrollment criteria have been described in detail previously[17,21].

**Sample collection.** Patient samples were collected as soon as possible after admission to the hospital. All samples were collected in combination with a clinical

blood draw. Blood for protein analysis was collected in a lithium heparin tube and separated into plasma and cell pellets. Components were frozen and stored in liquid nitrogen and -80 freezer respectively. Batch analysis was performed on plasma samples. Blood for flow cytometry was collected in a sodium heparin tube[20]. In patients who remained in hospital, samples were collected weekly.

Plasma from otherwise healthy control patients who were being evaluated for symptoms of a bleeding disorder (such as epistaxis) were obtained from discarded plasma from the coagulation lab at CHOP. A limited chart review was performed to confirm these patients had no comorbid medical illnesses. Patients found to have comorbid medical issues other than a possible isolated bleeding disorder were excluded. Patients who were eventually found to have a bleeding disorder were compared to patients who were not and no significant differences between groups were identified.

**Data collection.** Clinical and laboratory data were abstracted from electronic patient charts on to standardized case report forms created using the Research Electronic Data Capture database (REDCap version 11.2.2; Vanderbilt University, Nashville TN USA)[55]. Data were abstracted by a clinician or clinical research assistant. All data elements were validated by a physician.

**Clinical categorization.** Enrolled patients were prospectively classified in to 3 groups by physicians with expertise pediatric in hematology/oncology (CD, DT), pediatric infectious diseases (HB) or pediatric rheumatology (EB). Patient classification criteria have been described in detail previously[17]. In brief, Severe COVID-19 was defined as patients requiring new mechanical or non-invasive ventilatory support, or an increase in respiratory support above their baseline requirement. Minimal COVID-19 was defined as either an incidental finding of SARS-CoV-2 positivity during testing prior to an admission, diagnostic test or procedure or mild symptoms due to COVID-19 that did not require non-invasive mechanical ventilation. Patients who required low flow oxygen alone were categorized as Minimal COVID-19. MIS-C was defined per the Centers for Disease Control (CDC) criteria as fever, evidence of inflammation (elevated CRP, ESR or procalcitonin), multisystem organ involvement with at least two organ systems (cardiac, renal, respiratory, hematologic, dermatologic, gastrointestinal or neurologic), evidence of past or current positive SARS-CoV-2 infection by RT-PCR, serology or proven exposure to a close contact within four weeks prior to the onset of symptoms[56].

Patients were categorized as having thrombotic microangiopathy (TMA) based on criteria reported previously[17]. Only patients on whom a Hematoxylin-Eosin stained peripheral blood smear available were evaluated for TMA ($N = 34$). The criteria used included presence of thrombocytopenia, microangiopathic hemolytic anemia, and organ dysfunction. Hemolytic anemia was defined as anemia for age and schistocytes present on a peripheral blood smear. Organ dysfunction was defined as cardiac dysfunction (troponin greater than the upper limit of normal for age or requirement of inotropes), renal dysfunction (based on the Kidney Disease: Improving Global Outcomes (KDIGO) criteria)[57] or liver dysfunction (bilirubin greater than twice the upper limit of normal for age, alanine aminotransferase or aspartate aminotransferase greater than three times the upper limit of normal).

In patients on whom a ferritin was available ($N = 40$), a modified version of the Ravelli criteria for MAS were applied[32]. We modified the Ravelli criteria to exclude triglyceride levels because they were not measured in the majority of our patients. We defined a modified Ravelli criteria as ferritin > 684 ng/mL and any 2 of 1) platelet count <181,000/uL, AST > 48 U/L, or fibrinogen < 360 mg/dL. If any data were missing, they were imputed as negative/normal in order to bias our categorization towards the null hypothesis.

**Proteomic analysis.** Plasma protein levels were analyzed using the Olink Explore 1536/384 panel (Olink Proteomics, Uppsala, Sweden). Data were reported in

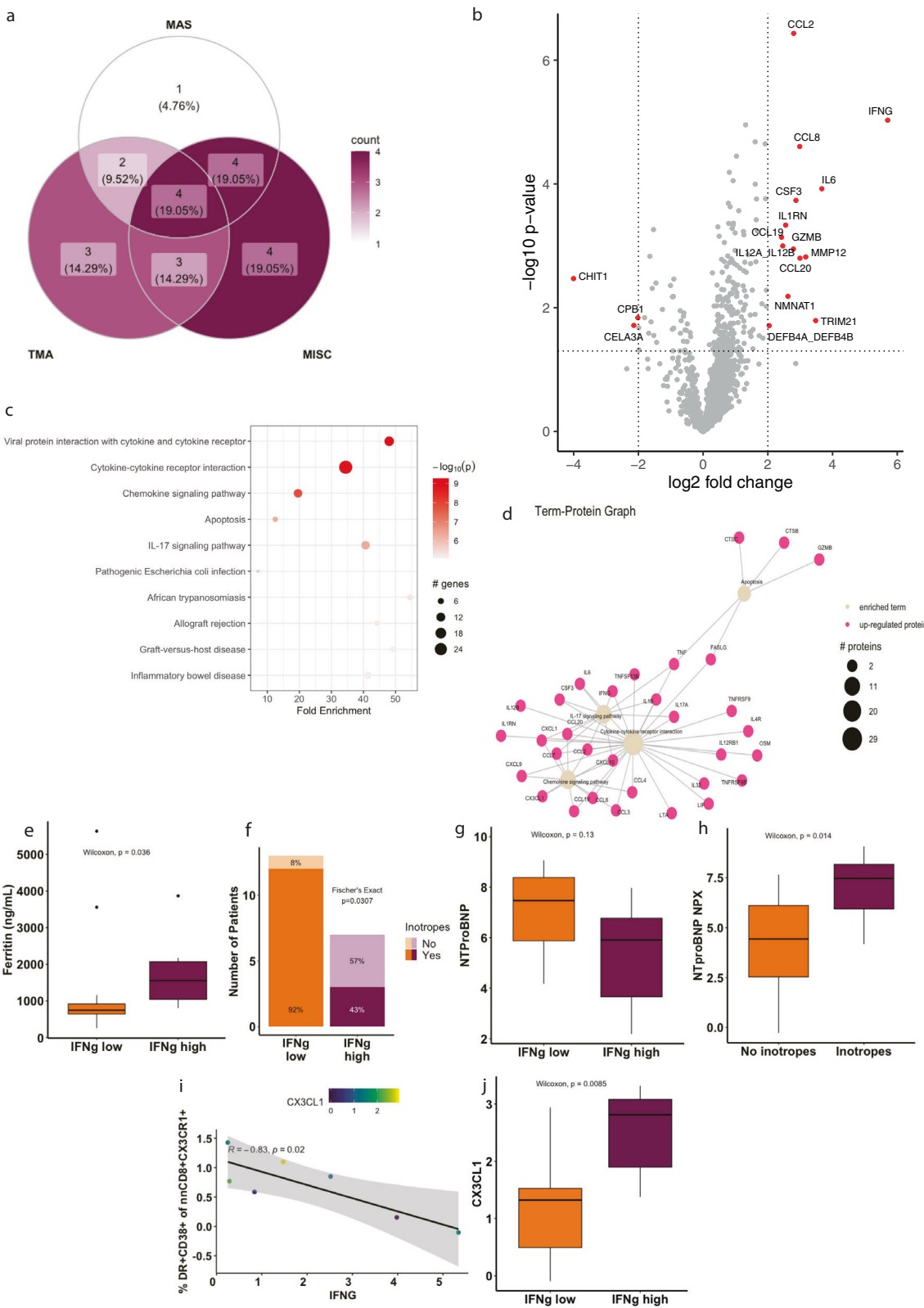

normalized protein expression values (NPX). NPX is an arbitrary unit in a Log2 scale calculated from inverted, normalized Ct values. All assay validation data are available on the Olink website ([www.olink.com](www.olink.com)).

**sC5b9 ELISA assay**. sC5b9 levels were determined using enzyme-linked immunosorbent assays (ELISA; Cat. #558315; BD Biosciences, San Jose CA, USA). Assays were performed at two dilutions; all samples were assayed in triplicate, using manufacturer protocols. As previously described, we set the upper limit of

normal for our sC5b9 assay at 247 ng/mL. sC5b9 measurements were row normalized to a Z-score and log2 transformed in order to be comparable to NPX scores.

**Flow cytometry analysis**. Detailed methods for flow cytometry analysis have been described previously[20]. Briefly, peripheral blood mononuclear cells (PBMCs) were isolated from blood collected in sodium heparin tubes. PBMCs were stained with a live/dead stain and then treated with Fc block before staining for chemokine

**Fig. 6 Clinical heterogeneity among MIS-C patients defined by their IFNγ signature. a** Overlaps between patients who meet criteria for macrophage activation syndrome (MAS), thrombotic microangiopathy (TMA), and MIS-C. **b** Differentially expressed proteins between patients with IFNγ-low ($N = 13$) and IFNγ-high expression ($N = 7$). Log2fold change threshold of 2 and a nominal $p$-value cutoff of 0.05 were used. **c** Unsupervised pathway ranking and **d** network analysis for differentially expressed proteins between IFNγ-low and IFNγ-high patients. **e** Maximum ferritin level during admission for patients in the IFNγ-low ($N = 12$) and IFNγ-high ($N = 7$) groups. $P$-value computed using Wilcoxon test. Horizontal line represents median, with bounds of box representing interquartile range. Whiskers represent 1.5x the interquartile range. Dots represent outliers. **f** Number of patients in IFNγ-low ($N = 13$) and IFNγ-high ($N = 7$) groups who did and did not require inotropic support. $P$-value computed with Fisher's exact test. NTproBNP expression between IFNγ-low ($N = 13$) and IFNγ-high groups ($N = 7$) (**g**) and in MIS-C patients who did ($N = 7$) and did not require inotropes (**h**; $N = 15$). Horizontal line represents median, with bounds of box representing interquartile range. Whiskers represent 1.5x the interquartile range. Dots represent outliers. **i** Correlation between IFNγ level and percent DR$^+$CD38$^+$ non-naïve CD8$^+$ CX3CR1$^+$ T cells in MIS-C patients ($N = 7$). Dots colored by CX3CL1 expression. $R$ value computed using Pearson's correlation coefficient after normality was demonstrated. Error band represents 95% confidence interval. **j** CX3CL1 levels between IFNγ-low ($N = 13$) and IFNγ-high groups ($N = 7$). $P$-values computed with Wilcoxon test. Horizontal line represents median, with bounds of box representing interquartile range. Whiskers represent 1.5x the interquartile range. Dots represent outliers. All $p$-values were calculated using two-sided tests. Source data are provided as a Source Data file.

receptors, surface markers, and intracellular markers. Samples were acquired on a 5 laser FACS Symphony A5 (BD Biosciences). All analysis was performed in FlowJo (Treestar, version 10.6.2). Flow cytometry data were transformed to a Z-score for all of the values for each gate used in order to be able to compare to the Log2 transformed NPX scores used in the Olink dataset.

**General statistical methods**. All statistical analyses were performed in R (version 4.0.4) using RStudio (RStudio, PBC, Boston, MA)[58]. Data were assumed to be non-parametric unless normality was demonstrated. Correlations were performed with Spearman's rank correlation coefficient for non-parametric data and Pearson's correlation coefficient for parametric data. Kruskall–Wallis testing was performed to compare three or more groups and Wilcoxcon signed-rank test was performed for paired groups. To test for associations between discrete variables, Fisher's exact test was used if small numbers were available in each cell. Otherwise, Chi-square testing was used. Unless otherwise stated, significance was based on a fold change greater than 2 or less the -2 and a false discovery rate (FDR; using Benjamini–Hochberg correction) value of less than 0.01.

**Clustering analysis**. The T-distributed Stochastic Neighbor Embedding (tSNE) for R package was used to perform tSNE clustering[59]. Principal component analysis was performed using R version 4.0.4. The factoextra package (https://cran.r-project.org/web/packages/factoextra/index.html) was used to extract and visualize PCA elements.

**Pathway analysis**. We completed a differential expression analysis of all proteins using a log2fold change threshold of 2 and FDR threshold of 0.01. Proteins from the Olink data set were inputted in to PathfindR along with their respective log2fold change and FDR value[25]. Using an active subnetwork enrichment analysis approach, PathfindR outputs a table that represents enriched pathways identified from the protein list inputted into the program. We used the KEGG pathway database (https://www.genome.jp/kegg/pathway.html) which includes a manual collection of pathway maps examining a total of 777,729 molecular pathways, with 544 main pathways included. Only pathways that had a $p$-value of $< = 0.01$ were considered. The final table produced by PathfindR includes a table of significant pathways with an associated adjusted $p$-value, a fold enrichment value of the pathway, the lowest and highest $p$-values generated from each iteration of the pathways analysis, and the upregulated and downregulated proteins from the input protein list for every pathway[25].

**Reporting summary**. Further information on research design is available in the Nature Research Reporting Summary linked to this article.

## Data availability
The NPX data generated in this study are provided in the Supplementary Information/ Source Data files. Data for Figs. 1 and 2 are provided in Source Data 1. Additional data for Fig. 3 is provided in Source Data 2. Additional data for Figs. 4 and 5 are provided in Source Data 3. Additional data for Fig. 6 is provided in Source Data 4. Additional Data contributing to Supplementary Fig. 1 is provided in Source Data 5. The Source Code is available in the Supplementary Methods in the Supplementary Information file. We also used the KEGG pathway database available here (https://www.genome.jp/kegg/pathway.html). Source data are provided with this paper.

## Code availability
R code used to generate the findings in this paper is provided in Supplementary Methods in the Supplementary Information file.

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

## Acknowledgements

We gratefully acknowledge the support of the Children's Hospital of Philadelphia Frontiers program and the assistance of Mr. Jansen Weaver, Mr. Daniel Fields and Mr. Kienan O'Brien. CHOP Frontiers Program Immune Dysregulation Team (DTT, EB, HB, KES, ML, SC), National Institute of Allergy and Infectious Diseases (NIAID): R01AI121250 (EMB), T32CA009140 (DAO), National Cancer Institute (NCI): R01CA193776 (DTT), X01HD100702-01 (DTT), 5UG1CA233249 (DTT), Leukemia and Lymphoma Society (DTT), Children's Oncology Group (DTT), Alex's Lemonade Stand Foundation for Childhood Cancer (DTT). HB is funded by Team Connor Childhood Cancer Foundation, Department of Defense Translational Team Science award CA170257, and NIH R61/R33 RADx-rad (Pre-VAIL kIds) 1R61DH105594. LAV is funded by a Mentored Clinical Scientist Career Development Award from NIAID/NIH (K08 AI136660). SC is funded by NIAID (R01 HD098428). EJW was supported by AI105343, AI082630, Allen Institute for Immunology, and the Parker Institute for Cancer Immunotherapy that supports the Cancer Immunology program at the University of Pennsylvania. CD was supported by the Institute for Translational Medicine and Therapeutics of the Perelman School of Medicine at the University of Pennsylvania and the Gail Slap Fellowship. This work was also supported by a generous donation from Jen and Fred Fox. JG was funded by T32 CA009140 and a Cancer Research Institute-Mark Foundation Fellowship.

## Author contributions

EMB, DTT, and HB equally contributed to this work and are co-senior authors. CD, SEH, LV, KES, MPL, EJW, EB, HB, and DTT designed the study. RS, FB, LV, JRG, AEB, DO, AF, CD, SC, and EMB analyzed data. CB, JRG, and AEB performed experiments. CD, JL, TL, and KOM enrolled subjects. CD, SH, KOM, JL, LV, TL, DTT, HB, and EMB abstracted clinical data. All authors contributed intellectually and reviewed and revised the manuscript.

## Competing interests

DTT serves on advisory boards for Janssen, Sobi, and BEAM. HB has stock ownership in Kriya Therapeutics. SH serves on the advisory board for Horizon Pharma. MPL is an advisory board member for Octapharma and Shionogi, a consultant for Amgen, Novartis, Shionogi, Dova, Bayer, the United States Department of Justice, Sobi, Principia and Argenx and has received research funding from Sysmex, Novartis, Astra Zeneca Rigel, Principia, Argenx, Janssen, and Dova, and has served as a medical advisor for Rigel, Principia, the Platelet Disorder Support Association, CdLS Foundation and 22q11.2 Society. KES received personal fees from Elsevier, Enzyvant and Immune Deficiency Foundation. HB is a paid consultant for Kriya Therapeutics. EMB receives research funding from AB2Bio. E.J.W. is a founder of Surface Oncology and Arsenal Biosciences. E.J.W. is an inventor on a patent (US patent number 10,370,446) submitted by Emory University that covers the use of PD-1 blockade to treat infections and cancer. The other authors declare no competing interests.
