## [Peer Review File · Nature Communications]

Proteomic Profiling of MIS-C Patients Reveals Heterogeneity Relating to Interferon Gamma Dysregulation and Vascular Endothelial DysfunctionREVIEWER COMMENTS

Reviewer #1 (Remarks to the Author):

The authors explore the plasma proteome of 22 children with MIS-C by using a multiplex, high-throughput protein biomarker platform that assesses a defined set of almost 1500 proteins. They compared this plasma proteome to that of children with mild (n=26) and severe (n=15) SARS-CoV-2 infection as well as healthy controls (n=25). They also included flow cytometric data assessing the distribution of lymphocyte subsets from 20 patients (MIS-C and COVID-19) that were reported in a previous study. The group of MIS-C clusters separately from the other groups using t-SNE analysis and showed some overlap with severe COVID-19 using PCA. The authors could replicate previous finding of high IFN- γ and IL-10 in MIS-C and provided new data suggesting a disproportionate IFN- γ /CXCL9 ratio in MIS-C. Patients that met the Ravelli MAS criteria (although not established for use in MIS-C, which was acknowledged by the authors) had higher IFN and CXCL9 levels. Interestingly, patients with high IFN- γ levels were less likely to require ionotropes. Additionally, they identified PLA2G2A as a biomarker that is differently expressed between all groups and highest in MIS-C. Patients meeting criteria for TMA were associated with significantly higher PLA2G2A levels and more likely to receive ionotropes.

Though being an explorative and descriptive study with a rather small sample size, which is acknowledged by the authors and is due to a rare pediatric disease, it provides very interesting and novel insights into the inflammatory proteome pattern in MIS-C that potentially correlate with distinct clinical phenotypes.

There are several points that I would like to address:

Major points:

The authors describe that the levels of PLA2G2A do not seem to change significantly between pre-treatment and early post treatment time points (p. 7, l149), "suggesting that sampling of plasma shortly after treatment does not immediately affect results". In my opinion, this generalized assumption seems questionable. I do understand that obtaining plasma samples from these rare patients is challenging, however, assessing the effect on treatment on the presented results in more detail would strengthen the manuscript. The authors should outline treatment (IVIg, steroid or both) at the time point of sampling in Supplemental table 1. The authors are also encouraged to highlight pre- and post-treatment samples in the t-SNE plots (Fig 1A) as well as the PCA (Fig 1E). This could for example be shown in a supplemental Figure.

The authors aimed at probing the source of IFN γ production by correlating IFN γ with soluble markers that might indicate T-cell (IL2RA), NK-cell (NCR1) as well as macrophage (CD163) activation (p8, lines 181-185). They could show positive correlation of IFN γ with IL2RA and NCR1 but not CD163. Further, they correlated IFN γ and CXCL9 levels with flow cytometric data obtained from these patients and reported in a previous study (p9, lines 195-197). CXCL9 but not IFN γ correlated with activated CD4 and CD8 cells. The authors concluded that these data suggests that "T cells are being activated by IFN γ rather than producing IFN γ ". In my view, these correlations are rather ambiguous and the conclusion that are drawn from this data are somewhat speculative and should be discussed cautiously. Also, the cytokines detected in the plasma could be derived from cells residing in affected tissues that have not been captured in this analysis.

The authors "investigated the intersections between the MAS and TMA phenotypes and MIS-C" (p12, l. 258). Are number of patients high enough to allow this analysis? Figure 4A shows the overlap of MAS, TMA and MIS-C. How do the patients that fall in the MAS-only (n=1, 4.76%) category differ from the ones that fall in the MAS + MIS-C category (n=4, 19.05%). Does it mean that the MAS-only do not have MIS-C? The same for TMA-only and TMA + MIS-C?

Minor points:

The colors used for MIS-C and Minimal are difficult to distinguish. The authors should consider a different set of colors.

For the healthy controls, patients that have a "possible isolated bleeding disorder" were used (p19, l403). Have bleeding disorders as far as possible eventually been excluded in these individuals? This seems particular of concern since coagulation alterations are part of MIS-C pathogenesis and

bleeding disorders in the control group could affect the results obtained in this study.

Reviewer #2 (Remarks to the Author):

The Authors analyzed 1400 proteins in patients affected with MIS-C and compared the proteomic profile with the severity of COVID-19 and flow cytometry extended panel of data.

Selecting the highly differentially expressed proteins, the authors were able to discriminate MIS-C pts from severe and minimal disease. Some overlap among severe and MIS-C was observed. A good response with return to base-line proteomic profile after treatment was observed, further supporting the strength of the data.

Of note, MIS-C patients had a disproportionately high CXCL9 response to IFN γ produced by activated T cells, resembling the hyper-inflammatory status of MAS. A subset of patients with MIS-C met the criteria for the IFN γ -associated MAS. In addition to CXCL9 response, in MIS-C patients, neutrophilia correlated strongly with CD163 but did not with IFN γ or CXCL9, implying that neutrophilia is associated with macrophage activation.

Phospholipase A2 was observed to be the most differentially expressed protein. A significantly higher PLA2G2A was observed in patients with MIS-C, who clustered. The Authors propose that PLA2G2A is a candidate biomarker for MIS-C.

For the first time an objective connection between IFN γ -high MIS-C patients and objective MAS features

What is surprising, and probably deserves further investigation is the observation that pts with high IFN γ expression appear to be protected from cardiac toxicity associated with MIS-C.

The data are absolutely convincing and fully supportive of the conclusion drawn. The data are illustrated in a proper form and all the figures are easy to be followed. The only criticism concerns the lack of correlation between high IFN γ and the surrogate marker of cardio toxicity. This part if possible should be mitigated.

I only have few minor comments:

In Figure 1A there are 2 different clusters for healthy patients. Is this due to different ages of other factors (i.e. different cell composition)? Please comment

The sentence on page 6 line 125-127 is too speculative since it is limited only to a few cases. I would suggest removing it.

In figure 1 I would suggest changing the colors of the dots referred to Severe and MIS-C since they are too similar and it is sometimes difficult to distinguish them especially when they are very close.

In figure 3 I would suggest avoiding the overlap between the symbols in the figure and the legend since the legend is not clearly visible

The paper deserves full consideration for publication.

Reviewer #3 (Remarks to the Author):

The goals of this study were to understand the pathophysiology of MIS-C via examination of the plasma proteome using OLINK platform, and to identify candidate biomarkers and predictors of disease severity. I think the OLINK data is of good quality, but data visualization and results text should be revised, it could be easily merged into one figure, table on DEPs and some suppl figures. Also the Results text should be considerably shortened and focus on the main biological insights.

Points to consider:

In Supplement M1 DEP 'prot names' are not protein names, but gene names. Protein names and UniProt entries should be provided. In addition, this table would be more informative if entries were sorted based on fold change. Also, the fold change should not be provided with up to eight decimals, I do not believe the quantification accuracy is on that level.

Fig 1: the PCA plots are not very informative, especially panels B-E which do not visualize the proteome data too well and could easily go into suppl info in most parts. Also the text on this in Results should be shortened considerably.

Fig2: volcano plots are fine, labelling of the proteins could be more clear (bigger fonts). However,

in the results text these are under 'pathways' even though volcano plots only visualize DEPs shown in Supplement M1. With pathway analysis I would again suggest showing only the most regulated pathways, now there are pathways that have hardly visible nodes, also not all the term-protein graphs shown are needed. I would just show the one for panel A in the main figure.

Like the authors state it is evident from Supplement M1 and volcano plots that PLA2G2A is much higher in patients than in controls, especially in MIS-C samples. The same is observed in Fig 5.

Overall I think most of the figures are partly complicated, the data would benefit from simpler presentation.

Whether the novelty of this study is enough to warrant for publication in Nature Communications is also something to consider. At present, the main findings are two candidate markers (IFN γ , PLA2G2A) for MIS-C and no functional insight for them is provided.

Reviewer #1

1. The authors explore the plasma proteome of 22 children with MIS-C by using a multiplex, high-throughput protein biomarker platform that assesses a defined set of almost 1500 proteins. They compared this plasma proteome to that of children with mild (n=26) and severe (n=15) SARS-CoV-2 infection as well as healthy controls (n=25). They also included flow cytometric data assessing the distribution of lymphocyte subsets from 20 patients (MIS-C and COVID-19) that were reported in a previous study. The group of MIS-C clusters separately from the other groups using t-SNE analysis and showed some overlap with severe COVID-19 using PCA. The authors could replicate previous finding of high IFN- γ and IL-10 in MIS-C and provided new data suggesting a disproportionate IFN- γ /CXCL9 ratio in MIS-C. Patients that met the Ravelli MAS criteria (although not established for use in MIS-C, which was acknowledged by the authors) had higher IFN and CXCL9 levels. Interestingly, patients with high IFN- γ levels were less likely to require ionotropes. Additionally, they identified PLA2G2A as a biomarker that is differently expressed between all groups and highest in MIS-C. Patients meeting criteria for TMA were associated with significantly higher PLA2G2A levels and more likely to receive ionotropes.

Response: We appreciate the reviewers thoughtful summary of our manuscript.

2. Though being an explorative and descriptive study with a rather small sample size, which is acknowledged by the authors and is due to a rare pediatric disease, it provides very interesting and novel insights into the inflammatory proteome pattern in MIS-C that potentially correlate with distinct clinical phenotypes.

Response: Thank you for the kind comments.

There are several points that I would like to address:

Major points:

3. The authors describe that the levels of PLA2G2A do not seem to change significantly between pre-treatment and early post treatment time points (p. 7, l149), “suggesting that sampling of plasma shortly after treatment does not immediately affect results”. In my opinion, this generalized assumption seems questionable. I do understand that obtaining plasma samples from these rare patients is challenging, however, assessing the effect on treatment on the presented results in more detail would strengthen the manuscript. The authors should outline treatment (IVIG, steroid or both) at the time point of sampling in Supplemental table 1.

Response: As suggested, we have amended Supplemental Table 1 to include further details about the treatment received at the timepoint of initial sampling for MIS-C patients. We agree that the comments about timing of sampling were overly generalized, and we have adjusted the paragraph in question referring to PLA2G2A. We have also expanded our commentary to refer to the decay in time seen with both PLA2G2A and IFN γ . We adjusted both figures to show pre- and post- treatment samples more clearly (PLA2G2A Supplemental Figure 1E included here

as an example). The paragraph now reads:

PLA2G2A pre- and post-treatment samples did not appear to differ significantly when measured in the acute time frame suggesting that sampling of plasma shortly after treatment does not immediately affect this analyte. Levels of PLA2G2A and IFN γ (Supplemental Figure S1E and S1F) did eventually decay over time following treatment.

Furthermore, in response to the reviewer's excellent comment below regarding highlighting pre- and post-treatment samples in the PCA plot (now Supplemental Figure 2F), we can see that pre- and post-treatment samples in MIS-C cluster together. We have added text describing this to page 7 of the manuscript:

We also examined the impact of treatment in all proteins in Supplemental Figure 2F. Samples drawn prior to or immediately after receiving treatment did not cluster separately from each other, implying minimal differences in the short-term following treatment.

4. The authors are also encouraged to highlight pre- and post-treatment samples in the t-SNE plots (Fig 1A) as well as the PCA (Fig 1E). This could for example be shown in a supplemental Figure.

Response: We agree with the reviewer that this is an important point. To show this important finding, we have made the suggested changes to Figure 1E (now Supplemental Figure 2E based on the suggestion of Reviewer 3), as well as Supplemental Figures 2A and 2F. As mentioned in the previous response, these figures demonstrate that there were not substantial differences between pre- and post-treatment samples in the acute timeframe.

The generation of tSNE mappings is a stochastic process. While the algorithm preserves local connections, the global geometry of tSNE space is different with every new analysis. Unfortunately, because we did not include patient-level IDs in the dataset to generate the tSNEs from the original submission, we cannot map pre-versus post-treatment status onto the tSNE maps from the original submission. To best preserve the integrity of the data visualization that the reviewers judged on the first submission, we felt it best to not regenerate a new tSNE mapping for the resubmission. However, per the request of the reviewer we did generate a new tSNE mapping to show pre- and post-treatment status shown here. As in the PCA, there is no clear separation of pre-versus post-treatment samples during the acute time frame.

5. The authors aimed at probing the source of IFN γ production by correlating IFN γ with soluble markers that might indicate T-cell (IL2RA), NK-cell (NCR1) as well as macrophage (CD163) activation p8, lines 181-185). They could show positive correlation of IFN γ with IL2RA and NCR1 but not CD163. Further, they correlated IFN γ and CXCL9 levels with flow cytometric data obtained from these patients and reported in a previous study (p9, lines 195-197). CXCL9 but not IFN γ correlated with activated CD4 and CD8 cells. The authors concluded that these data suggests that “T

cells are being activated by IFN γ rather than producing IFN γ ". In my view, these correlations are rather ambiguous and the conclusion that are drawn from this data are somewhat speculative and should be discussed cautiously. Also, the cytokines detected in the plasma could be derived from cells residing in affected tissues that have not been captured in this analysis.

Response: We have removed the speculative wording about the source of interferon gamma from the results section on page 9.

6. The authors "investigated the intersections between the MAS and TMA phenotypes and MIS-C" (p12, l. 258). Are number of patients high enough to allow this analysis? Figure 4A shows the overlap of MAS, TMA and MISC. How do the patients that fall in the MAS-only (n=1, 4.76%) category differ from the ones that fall in the MAS + MISC category (n=4, 19.05%). Does it mean that the MAS-only do not have MIS-C? The same for TMA-only and TMA + MISC?

This is an excellent point. We have added an acknowledgement of the small sample size to the paragraph in question on page 12, line 268. We have also clarified the wording around this analysis in response to the reviewer's comment. The paragraph now reads:

Next, we investigated the intersection between the MAS and TMA phenotypes and MIS-C. In all patients who had available data for analysis of MAS and TMA, we looked at overlap with MIS-C (N=21). Among MIS-C patients (N=15), TMA and MAS occurred independently (Fisher's exact, $p = n.s.$; Figure 6A), although the sample size was small. When TMA and MAS phenotypes were overlayed on the same tSNE mapping used in Figure 1, patient clustering by disease state was evident (Supplemental Figure S4A).

Minor points:

7. The colors used for MIS-C and Minimal are difficult to distinguish. The authors should consider a different set of colors.

Response: We have changed the color schemes across figures.

8. For the healthy controls, patients that have a "possible isolated bleeding disorder" were used (p19, l403). Have bleeding disorders as far as possible eventually been excluded in these individuals? This seems particular of concern since coagulation alterations are part of MIS-C pathogenesis and bleeding disorders in the control group could affect the results obtained in this study.

Response: Normal controls were collected using discarded serum in patients with coagulation studies sent. Patients with significant pre-existing medical conditions were excluded. Some of the children were eventually diagnosed with a common mild bleeding disorder, such as von Willebrand disease. To address the reviewers concern, we compared the patients who were subsequently diagnosed with a mild bleeding disorder with those who had a negative work-up. No differences were identified comparing these two cohorts. Shown below is a principal component analysis comparing patients with and without bleeding disorders.

We have added a line to the manuscript on page 19 to confirm that there were no differences between patients with and without bleeding disorders.

Reviewer #2

1. The Authors analyzed 1400 proteins in patients affected with MIS-C and compared the proteomic profile with the severity of COVID-19 and flow cytometry extended panel of data.

Selecting the highly differentially expressed proteins, the authors were able to discriminate MIS-C pts from severe and minimal disease. Some overlap among severe and MIS-C was observed. A good response with return to base-line proteomic profile after treatment was observed, further supporting the strength of the data.

Of note, MIS-C patients had a disproportionately high CXCL9 response to IFN γ produced by activated T cells, resembling the hyper-inflammatory status of MAS. A subset of patients with MIS-C met the criteria for the IFN γ -associated MAS. In addition to CXCL9 response, in MIS-C patients, neutrophilia correlated strongly with CD163 but did not with IFN γ or CXCL9, implying that neutrophilia is associated with macrophage activation.

Phospholipase A2 was observed to be the most differentially expressed protein. A significantly higher PLA2G2A was observed in patients with MIS-C, who clustered. The Authors propose that PLA2G2A is a candidate biomarker for MIS-C.

For the first time an objective connection between IFN γ -high MIS-C patients and objective MAS features

What is surprising, and probably deserves further investigation is the observation that pts with high IFN γ expression appear to be protected from cardiac toxicity associated with MIS-C.

The data are absolutely convincing and fully supportive of the conclusion drawn. The data are illustrated in a proper form and all the figures are easy to be followed. The only criticism concerns the lack of correlation between high IFN γ and the surrogate marker of cardio toxicity. This part if possible should be mitigated.

Response: We appreciate the reviewer's detailed reading of our manuscript and their kind comments about our manuscript. With respect to their comment about high IFN γ and the NTproBNP we believe that the trend approaches significance, however, we were underpowered to detect a difference.

I only have few minor comments:

2. In Figure 1A there are 2 different clusters for healthy patients. Is this due to different ages of other factors (i.e. different cell composition)? Please comment
3. *Response: We agree that that there are 2 different clusters for healthy patients on the tSNE map. We note that in tSNE space, distance between points is not linearly correlated with the differences between them. In PCA space (now shown in Supp. Fig. 2A), we also note that there is the slight appearance of two clusters of healthy as well. However, the differences between these two clusters appears much smaller, reflecting that in PCA space, distance is linearly correlated to differences, and therefore the difference between the two healthy clusters is not as large as the distortion of tSNE space makes it seem. Nonetheless, we considered how these two clusters may differ from each other. We analyzed these two different clusters for differences between*

age, gender and ethnicity and there were no significant differences between the two groups. As mentioned in response to point 8 from Reviewer 1, there were also no differences in diagnosis of mild bleeding disorder. Therefore, we were unable to identify what the nature of these two clusters is, but none of the measurable confounders in our dataset account for these differences. We note that in the major analytes we consider in the latter part of the manuscript, there are no differences between these two clusters of healthy controls, thus whatever differences are driving their segregation do not seem to contribute to the results of these analyses.

4. The sentence on page 6 line 125-127 is too speculative since it is limited only to a few cases. I would suggest removing it.

Response: We have removed the sentence per the reviewer's suggestion.

5. In figure 1 I would suggest changing the colors of the dots referred to Severe and MIS-C since they are too similar and it is sometimes difficult to distinguish them especially when they are very close.

Response: We have changed the colors of the dots.

6. In figure 3 I would suggest avoiding the overlap between the symbols in the figure and the legend since the legend is not clearly visible

Response: We have reformatted the figures and changed the legend colors. We believe the legends are now more clearly visible.

7. The paper deserves full consideration for publication.

Response: We appreciate the reviewer's kind comment.

Reviewer #3

1. The goals of this study were to understand the pathophysiology of MIS-C via examination of the plasma proteome using OLINK platform, and to identify candidate biomarkers and predictors of disease severity. I think the OLINK data is of good quality, but data visualization and results text should be revised, it could be easily merged into one figure, table on DEPs and some suppl figures. Also the Results text should be considerably shortened and focus on the main biological insights.

Response: We appreciate the thoughtful comments and have significantly condensed the figures as the reviewer suggested below. We have also considerably shortened the results text and believe the manuscript reads more clearly following these changes.

Points to consider:

2. In Supplement M1 DEP 'prot names' are not protein names, but gene names. Protein names and UniProt entries should be provided. In addition, this table would be more informative if entries were sorted based on fold change. Also, the fold change should not be provided with up to eight decimals, I do not believe the quantification accuracy is on that level.

Response: We have made the suggested changes to Supplement M1. We have changed the heading of Supplement M1 to Protein Names and provided UniProt entries. Entries have been sorted based on fold change. Fold-changes are no longer

reported with eight decimals.

3. Fig 1: the PCA plots are not very informative, especially panels B-E which do not visualize the proteome data too well and could easily go into suppl info in most parts. Also the text on this in Results should be shortened considerably.

Response: We have moved the PCA plots to Supplemental Figure 2. To shorten the section in the results, we also removed Figure 1C and the text describing it altogether.

4. Fig2: volcano plots are fine, labelling of the proteins could be more clear (bigger fonts). However, in the results text these are under 'pathways' even though volcano plots only visualize DEPs shown in Supplement M1. With pathway analysis I would again suggest showing only the most regulated pathways, now there are pathways that have hardly visible nodes, also not all the term-protein graphs shown are needed. I would just show the one for panel A in the main figure.

Response: We appreciate the reviewers careful reading of our manuscript. We have removed the reference to "pathways of activation" in reference to the Volcano Plots from Page 7 line 169.

We have reduced the pathway analysis in Figure 2 to show only the top 5 pathways, instead of the top 10 to avoid showing pathways that are not regulated. We have removed the term-protein graphs in Figure 2B and 2C as suggested by the reviewer.

5. Like the authors state it is evident from Supplement M1 and volcano plots that PLA2G2A is much higher in patients than in controls, especially in MIS-C samples. The same is observed in Fig 5.

Response: We agree with the reviewer that PLA2G2A is significantly higher.

6. Overall I think most of the figures are partly complicated, the data would benefit from simpler presentation.

Response: We have made all suggested changes to the figures and believe that the new configuration more clearly represents the data.

7. Whether the novelty of this study is enough to warrant for publication in Nature Communications is also something to consider. At present, the main findings are two candidate markers (IFN γ , PLA2G2A) for MIS-C and no functional insight for them is provided.

Response: We believe that we present in depth work on a robust number of pediatric patients with a rare condition. The presentation of the proteomic changes in this condition is an important first step in understanding how to design hypothesis driven research to better understand the mechanisms underpinning MIS-C. We hope the Reviewer will agree that the identification of the candidate biomarkers, contextualized with the additional clinical and proteomic data supporting their relevance beyond the analysis of a single analyte is an important contribution. We believe that this work will serve as an important foundation for further mechanistic studies in the future.

REVIEWERS' COMMENTS

Reviewer #1 (Remarks to the Author):

Thank you for addressing all points that I have raised. I do not have any further concerns. The study is an important contribution to the field.

Reviewer #2 (Remarks to the Author):

The authors modified the paper according to the suggestions and it is now suitable for publication.

Reviewer #3 (Remarks to the Author):

i think the authors have addressed my comments from the previous review

Response to Reviewer Comments

REVIEWERS' COMMENTS

Reviewer #1 (Remarks to the Author):

Thank you for addressing all points that I have raised. I do not have any further concerns. The study is an important contribution to the field.

We appreciate the reviewers kind remarks.

Reviewer #2 (Remarks to the Author):

The authors modified the paper according to the suggestions and it is now suitable for publication.

We appreciate that the reviewer finds our manuscript suitable for publication.

Reviewer #3 (Remarks to the Author):

i think the authors have addressed my comments from the previous review

We are glad that the reviewer feels we have sufficiently addressed their comments.